# Influence of Mineralized Water Sources on the Properties of Calcisol and Yield of Wheat (*Triticum aestivum* L.)

**DOI:** 10.3390/plants11233291

**Published:** 2022-11-29

**Authors:** Evgeny Abakumov, Gulomjon Yuldashev, Dilmurod Darmonov, Avazbek Turdaliev, Kamoliddin Askarov, Mavlonjon Khaydarov, Ulugbek Mirzayev, Timur Nizamutdinov, Kakhramonjon Davronov

**Affiliations:** 1Department of Applied Ecology, Faculty of Biology, Saint Petersburg State University, 16 Line of V.O., 199178 Saint-Petersburg, Russia; 2Fergana State University, Fergana 150100, Uzbekistan; 3Fergana Polytechnic Institute, Fergana 150107, Uzbekistan

**Keywords:** melioration, irrigation, soil fertility, agrosoils, calcisols, Uzbekistan

## Abstract

The use of conservation agriculture (SWC—soil and water conservation) technologies is now becoming more and more necessary. For the soils in arid ecosystems, the problem of irrigation deficiencies has always been relevant, and clean fresh water is always insufficient to irrigate these agricultural lands. This paper provides a brief historical overview of the use of mineralized water sources in agriculture and their impacts on soils and plants (*Triticum aestivum* L.). The experiment involving wheat cultivation in saline soils irrigated with mineralized water was set for 3 years. The main chemical and physical–chemical properties of the agro-transformed solonchaks and mineralized water sources were investigated. According to the contents of mobile forms of N, P, and K, the soils were poorly supplied; after a series of irrigation phases, they remained the same. There were signs of the growth of mobile phosphorus in the variants where mineralized water sources were applied. Our results showed that under conditions of irrigation with water sources with mineralization rates of up to 2.8–3.5 g/L, the wheat yield increased by 1.5 c/ha compared to the control. The use of mineralized water for irrigation purposes will reduce the use of clean river water.

## 1. Introduction

Today, around the world, 50% of the drinking water and 43% of the irrigated water originates from soil and groundwater. About 20% of the total agricultural land areas are irrigated [1]. In the last 50 years, the area under crops around the world has grown by 12%, and annually about 300 km^3^ of collector–drainage water is formed around the world, including in the NIS countries (90 km^3^) and the USA (30 km^3^). As a result of the global growth in production in some regions and the growing degradation of land and water resources, the ecological state of soils and water reserves is deteriorating. Recently, it was shown that the use of integrated SWC for 5 years increased grain yields by 72.8% over the control [1].

For this reason, it is necessary to determine the impact of SWC on agricultural soil fertility and its chemical and geochemical properties under the conditions of irrigation with mineralized water. Irrigation with mineralized water may help to increase grain yields and to use land and water resources more sustainably. Around the globe, the need for water is constantly growing due to the growth and development of national economies. As well as the increasing grain yields and the efficient use of land and water resources, at the same time the use of mineralized water on the spot combined with education is of great theoretical and practical importance. Mineralized water represents a huge source of water for irrigation, especially in the semi-arid and arid regions of Eurasia. These water sources could be of various origins, including natural and secondary sources. Natural water sources are mineralized due to salt accumulation in the rivers from adjacent landscapes. Secondary water sources circulate in irrigated landscapes and come into contact with salinized soils several times.

Numerous scientists have studied soil transformation and the dynamics of crop yields both in the NIS countries and around the whole world [2,3]. In 1903, Means, having studied irrigation in North Africa, concluded that when using drainage, highly mineralized water sources can be successfully used, but in non-drainage conditions, when irrigated with even slightly mineralized water sources, the soil can be removed from agricultural circulation. When irrigated with mineralized water sources, Arab peasants received high yields of palm fruits and horticultural crops. The mineralization rate in this case was 8 g/L, and the amount of sodium chloride reached 50% of the total salts. On light, well-permeable soils, they applied water at concentrations of up to 10 g/L. In California, beginning in 1910, an orange orchard was watered with highly mineralized water. The soil and subsoil from this garden were light and permeable, with the groundwater level being deeper than 45 m from the ground. At a depth of 12 m, the conditional concentration of the soil solution at the full-field moisture capacity of the soil was 3 times lower than the concentration of water used for irrigation, and for 30 years the garden developed normally and bore fruit [3]. Kelly recommends using water with a chlorine content range of no more than 0.15–0.20 g/L for light California soils. The plant height is a genetic characteristic of a variety, but its potential can be achieved via adequate crop management [4].

According to Legostaev in 1932, irrigating wheat in the conditions of the Volga region with water sources with a mineralization range of 3.65–4.57 g/L, including chlorine over a range of 1.17–1.48 g/L, resulted in intense soil salinization [5]. It should be noted that theoretically, with the intensive leaching of salts from the weakly gypsum-bearing solonchaks of Central Asia, the temporary solonetzization of soils may occur. Antipov-Karataev [6], studying the physicochemical properties of soils depending on the composition and ratio of exchangeable cations, noted that the replacement of Ca or Mg separately or Ca and Mg simultaneously in the soil-absorbing complex with sodium is accompanied by an increase in soil dispersion. For example, the partial replacement of adsorbed calcium with magnesium increases the dispersion over a content range of 25–50% magnesium due to the exchange capacity, which can be caused by the addition of 5% exchangeable sodium. In this regard, the focus shifts to the mandatory accounting of irrigation water, together with other components of the Na cations, bearing in mind that it enters, depending on a number of factors, into the soil-absorbing complex and imparts unfavorable properties to the soil.

Kovda points out that in the salt regime of soils, irrigation water sources, which contain salts, are of great importance [7]. This is why in addition to the amount of irrigation water, it is also necessary to take into account its mineralization. He gives examples where in the Fergana Valley and the Hungry Steppe, as in many irrigated areas of southern Ukraine, water sources containing salts in the range of 4–6 g/L were used. The fate of the salts with such an input develops in different ways, depending on the hydrogeology, geomorphology, and drainage of the territory. Novikoff reports that in Tunisia and Iraq, farmers have been irrigating the land with mineralized water sources with dense residue rates of up to 5 g/L for centuries [8]. Kostyakov states that the allowable salt content range for plants in irrigation water is from 0.10 to 0.15%, with the salt concentrations corresponding to 1000 kg/1000 m^3^ of water [9]. He believes that the harmful effect of the salts depends on the nature of the soil. In well-permeable soils, the salt contents are to be allowed no higher than Na_2_CO_3_ > 0.1%, NaCI > 0.2%, and Na_2_SO_4_ > 0.5%, and even less for the sum of salts. If the water contains a lot of gypsum then it is harmless. If the composition contains Na_2_CO_3_ > 0.1%, then such water is unsuitable, while if it contains NaCl and Na_2_SO_4_, it is suitable only for light soils. He notes that the irrigation water may have a low concentration of salts, but with a large amount of ions that are harmful to plants. Such water is completely unsuitable for irrigation.

Kovda provided the US classification, which indicates the main properties of the water that determine its irrigation qualities. This classification takes into account the following:The total concentration of soluble salts;The Na concentration and the ratio of Na to the sum of Ca and Mg;The amount of bicarbonates;The concentrations of boron and other substances in toxic amounts [10].

At present, it is well known that if the Na ions predominate in the irrigation water, the latter will tend to displace Ca and Mg from the absorbing complex. To speed up the melioration of solonetzes, gypsum should be added to the irrigation water if this water does not contain easily soluble calcium salts. The Moroccan Agronomic Laboratory in Casablanca gives the following scale for assessing mineralized water sources in terms of the chloride content expressed in terms of NaCl:-0.5 g/L—suitable for the irrigation of any crop;-0.5–1 g/L—suitable for the irrigation of most crops;-1–1.5 g/L—suitable for irrigation, except for slightly salt-tolerant crops;-1.5–2 g/L—suitable for the irrigation of most crops, with the exception of low salt tolerance crops, with appropriate agricultural technology and soil drainage;-2–3 g/L—suitable for the irrigation of some crops, subject to appropriate precautionary rules;-3–4 g/L—practically unsuitable for irrigation;-4.0 g/L—unsuitable for irrigation.

It is noted that the richer the soils with organic matter and gypsum, the greater the permissible mineralization of the irrigation water. The indicated scale unfortunately does not take into account the soil conditions or the composition of salts in the soil profile (except for NaCl). Irrigation with mineralized water can lead to adverse effects at the expense of other salts. After all, water contains not only NaCl.

Despite the indicated changes in soils under the influence of irrigation with mineralized water sources and the irrigation regime, as well as the agrochemical and biogeochemical characteristics of the irrigated soils, this problem has been given great attention by a number of scientists [11,12].

The irrigation of agricultural plants, regardless of the type of soil, largely depends on the mineralization and chemical composition of the irrigation water. Irrigation with groundwater and the construction of vertical drainage areas for land reclamation have become widespread practices in many countries around the world. In India more than 5 million ha and in China more than 2.5 million ha are irrigated with groundwater, which is mineralized to one degree or another.

Speaking about the possibility of using collector–drainage mineralized water for irrigation, we can say that this leads not only to the expansion of irrigated areas, but at the same time partially solves the problem of the radical reclamation of large areas of saline land.

Bespalov conducted a series of experiments on the practical use of mineralized water sources in gray soils and meadow soils, and gave the following recommendations. In gypsum soils, mineralized groundwater does not create conditions for the formation of a solonetz process, even if the mineral present in these water sources is chloride. In soils not containing gypsum that are mineralized with chloride or sodium sulfate, the water can be used for irrigation only if the mineralization rate does not exceed 0.25 g/L. The authors found that using mineralized water with concentrations of up to 6 g/L on light, well-permeable soils with increased irrigation rates did not reduce the yield of raw cotton. On the soils of the Chardjou and Khorezm experimental stations, when using water with a dense residue range of 5–7 g/L, a significant decrease in the yield of raw cotton was observed [13].

The ratio of anions and cations in irrigation water can have different effects on plants under different hydrogeological and natural conditions. When irrigating plants with collector–drainage water, it is necessary to ensure that the concentration of the soil solution is not higher than the limits at which the plants are able to develop normally. It is necessary to create other conditions to obtain a high cotton yield.

Antipov-Karataev revealed that under the influence of the alkalinity of the medium, the absorption capacity of the soil increases as a result of the inclusion of ionogenic groups (similar to polyfunctional ones) in the reaction, which in turn leads to the additional absorption of sodium [6].

It was shown that saline water reduced the grain yield ratios by 8.5%, 11.0%, and 9.7% compared to non-saline water during the 2019/2020 and 2020/2021 seasons and over both seasons, respectively, in the soils of the Experimental Farm of Shandaweel Agricultural Research Station, Agricultural Research Center (ARC), Sohag, Egypt [10].

Therefore, against a background of soda solutions, soil alkalinity very quickly occurs, even with low degrees of mineralization in the irrigation water. An approximate calculation of the permissible critical amount of exchangeable Na can be carried out according to the content of soda in water, assuming that approximately 50% of its sodium will be absorbed by the soil in the first years of irrigation with such water. Improving the quality of such water should be carried out by adding calcium salts to it or by adding gypsum to the soil surface in advance, i.e., before water is supplied, which is very important for neutralizing the soda.

It should be noted that the Ferghana Valley, like some other regions of Uzbekistan, experiences acute shortages of irrigation water, especially in dry years. The total water supply range in the Fergana region is 60–70%, meaning collector–drainage artesian water sources are increasingly being used, which makes it possible to partially improve the water supply. Bespalov, summing up the research work on the pressure advice system, cited that in the Khorezm and Fergana regions, the irrigation of crops with groundwater should be alternated with irrigation with river water [13]. When using highly mineralized water sources for vegetative irrigation, the possibility of soil salinization increases; it is recommended that water with a mineralization range of 3–4 g/L be used for the irrigation of agricultural crops. Kiseleva found that the irrigation of cotton with mineralized water in the Hungry Steppe reduced the yield [14,15]. Moreover, the crop residues that were returned into the soil during conventional tillage resulted in a lower soil bulk density range in the deep soil layers (20–60 cm) [16].

Scientists from the Kuban region conducting drip irrigation experiments using saline collector–drainage water sources have achieved essential progress in this area [17]. Kruger watered cotton in vessels with artificially created water sources with the following compositions: 1—NaCl = 0.58 g/L; 2—NaCl = 1.46 g/L; 3—NaCl = 2.92 g/L. The authors found that in typical gray soil, which was non-saline and devoid of gypsum, the growth and development of the cotton depended mainly on the content of NaCl in the irrigation water. With an increase in NaCl in the water, the yield decreased, which was associated with an increase in salinity [18].

Vladychensky notes that water sources with salinity above 5 g/L are unacceptable for irrigation. Vladychensky also noted, in the case of a solid residue content range in water of 1–5 g/L, the chemical composition of the salts, the nature of the soil, and the salt tolerance of the plants should be taken into account. The author gave a series of toxicity (Table 1) ranges for sodium salts, taking the toxicity of Na_2_SO_4_ as the measurement unit [17].

In recent years, similar studies have been conducted by numerous researchers, but there are practically no studies related to the irrigation of wheat [18,19,20,21,22,23,24,25,26,27,28,29,30,31,32].

The purpose of this study is to determine the soil alterations under the influence of the irrigation of wheat plantations in the saz medium loamy soils of the Fergana Valley with water sources with various degrees of mineralization.

***Research objectives***:-To determine the degree of mineralization in the water sources regarding the danger of the salinization and alkalinization of the irrigated soils;-To establish local water quality limits for wheat, taking into account the properties of the soil, which could be the basis for the draft standard, “Nature Protection, Hydrosphere”;-To study the changes in the physicochemical properties of irrigated meadow saz soils under the influence of mineralized water sources of various concentrations and compositions;-To study the influence of the permissible mineralization of ditch and mineralized water sources of various compositions and concentrations on the growth, development, and yield of wheat.

## 2. Materials and Methods

In order to study the effects of irrigation with mineralized water sources on the properties of medium loamy meadow soils and the “Polovchanka” wheat variety (*Triticum aestivum* L.) in 2016 and 2018, field experiments were carried out. The soil study was conducted based on one soil monitoring section. The sampling was conducted on each soil genetic horizon in three replications, and the mass of each individual soil sample was about 100 g. Irrigation water samples were collected in chemically inert plastic bottles and immediately transported to the chemical laboratory.

***Description of the study site***. As the focus of study in 2012, irrigated meadow saz medium loamy soils, which are common in the territory of the Kuva district on the border of the deserts and the serozem soil belt (Calcisol) of the Fergana region, were selected. The experimental plot was located on the arable irrigated field in central part of Fergana valley.

The wheat variety “Polovchanka” was chosen for sowing. The area was limited from 3 sides by open-type drains and only from the north by the drainage water collector. The total area measured 11.5 hectares (Figure 1).

The experiments were carried out with three repetitions and 4 treatments in one tier. The area of each treatment measured 224 m^2^ (5.6 × 40). The experiments were carried out according to the “Methods of Agrophysical Research” [33] and “Methods of Agrochemical Research of Soils and Plants” [34] guidelines. The mathematical and statistical processing was carried out according to the methodology used by Karimov and Yuldashev [35]. The statistical processing of the data was performed using multivariate and univariate analyses of variance (MANOVA and ANOVA), as well as a correlation analysis. The variants of the water treatment with different degrees of salinity were chosen as the resultant factors for the analysis of variance; different chemical and physical–chemical characteristics of the soils were chosen as the dependent variables. The analysis was carried out using Statistica v12.0 and GraphPad Prizm v9.0.0 software.

The soil samples were taken in the spring and fall seasons of 2016 and 2018 from the next area and the field experiments:

Treatment 1. Irrigation with ditch (irrigational channel; this channel is supplied by water from the river) water at a rate of 800 m^3^/ha;

Treatment 2. Irrigation with drainage mineralized water at a rate of 800 m^3^/ha;

Treatment 3. Irrigation from the collector of primary used water in the merge area of the field at a rate of 800 m^3^/ha;

Treatment 4. Irrigation with mixed water at a rate of 800 m^3^/ha.

In all treatments in the experiment, wheat of the Polovchanka variety was sown for 3 years in a row. Phenological observations were carried out every 2 weeks. In all variants, in accordance with the design of the study, samples of the plow (arable) and subplow horizons were taken further along the genetic horizons to the groundwater level. In addition, samples of the canal and ground, drainage, and collector–drainage water were taken during irrigation.

The laboratory studies of selected soil and water samples included the following definitions:(a)The structural state of soils according to Savinov [33];(b)The humus content according to Tyurin’s method [34];(c)The gross nitrogen, phosphorus, and potassium contents in one sample according to the methods used by Maltseva and Gritsenko [33];(d)The mobile phosphorus content according to the Machigin method [33];(e)The mobile potassium content from 1% carbon ammonium extract according to Protasov [33];(f)The water-soluble salt content using the water extracts method [33];(g)The mechanical composition of the soil using a pipette method using sodium hexametaphosphate [33];(h)The content of absorbed bases using the Pfeffer method modified by Kruger and Queen [33].

## 3. Results and Discussion

The ratio of cations and anions in mineralized irrigation water can have different effects on the growth, development, and yield of the plants in different soil and climatic conditions in different ways. When irrigating plants with mineralized water sources, it is necessary to conduct monitoring studies of the salt composition, absorption capacity of the soils, the level of alkalinity, and other soil properties. Based on the above, we carried out a series of experiments according to the scheme shown in Table 2 for the retention of dry residues, as well as the ratio of anions and cations in the conditions of the deserts of Central Fergana, while the irrigation of Polovchanka wheat plants was carried out taking into account the irrigation regime.

To characterize the soils of the study area, a total of 24 soil sections and 8 pits were laid out on the territory of the farm in the Kuva district of the Fergana region. We present a morphological description of the most typical profiles below.

The soil sections described in the field in 2016 were collected in the spring, before irrigation with mineralized water sources, at a distance of 80 m from the drainage water collector in the south direction. The typical soils in the study plot were previously irrigated saz meadow (Calcisol (WRB 2015)) soils with a medium loamy mechanical composition.

Ap1. 0–32 cm: The arable layer was gray, slightly moist, medium loamy, slightly dense, and lumpy. Single roots and half-decayed plant remains of reeds were found, and there were small passages of shrews, the transition was sharp in density.

Ap2. 32–50 cm: The arable layer was light gray with darkish hues, slightly moist, medium loam, dense, and lumpy, with rare roots and passages of shrews and no exclusion. The pedological features were in the form of small gypsum crystals, with transitions in color and density.

B1. 50–70 cm: The arable layer was light gray with bluish spots and moist, medium and heavy loamy, and dense. No shrew burrows or root residues were found. There were accumulations of gypsum and water-soluble salts, and the transition was constant in color.

B2. 70–90 cm: The arable layer was gray with reddish-gray spots and was moist, medium loamy, dense, and crisp, containing a lot of gypsum and carbonate. There were small gypsum crystals.

BCA. 90–110 cm: The arable layer was bluish-red, wet, medium loamy, and dense, containing a lot of gypsum, with a clear transition in color.

From 110–130 cm: The arable layer was reddish, wet, medium loamy, and dense-crunchy, containing a lot of gypsum from 110 cm and below, and the ground water tasted fresh.

The soil investigated in the soil section was typical of the dry subtropical climate of the central part of the Fergana Valley. These soils formed on ancient soil-forming rocks of proluvial and alluvial origin. Since the central part of the Fergana Valley is an accumulative landscape, all parent materials accumulate barely soluble salts (gypsum, calcite) and easily soluble salts of various compositions

As expected, for such a short period (3 years) under the influence of irrigation activity, no significant changes were observed in the morphological characteristics of the soils. The groundwater was found at depths of 130–150 cm and did not seem to be mineralized. These relatively small morphological changes are associated with the mechanical and structural composition of the studied soils. The changes in the particle size distribution of the soils under the influence of irrigation with river and mineralized water sources are shown in Table 2.

From the data given in the table, it can be seen that the soils before and after wheat irrigation remained medium loamy in terms of texture. At the same time, the amounts of physical clay fluctuated in the soils before irrigation with mineralized water sources, i.e., in fall 2016, in the range of 30.5–40.4%. At the same time, the content of physical clay was higher in the arable horizons, at 40.4%, which was associated with the beginning of the formation of an agro-irrigation horizon. Further to this indicator, the content of physical clay was practically the same throughout the soil profile and varies within the range of 30.5–32.6%. As expected, the largest content of particles with a size range of 0.05–0.01 mm of coarse dust is characteristic of these medium loamy soils. The presence of this fraction in the soil-forming rocks and soils is a sign of the aeolian accumulation of coarse dust particles, which indicates the loess origin of the rocks. The content of this fraction varied in the studied soils in the range of 37.8–48.5%. The content range of macroaggregates in arable horizons is low and amounts to 0.70–0.72%. The effect of irrigation with mineralized collector and drainage water sources on the mechanical composition of medium loamy soils, i.e., significant changes in the contents of different fractions, did not occur. However, there was a slight change in the content of the physical clay.

For example, the content of physical clay in the plow horizon of the original soils was 40.35%, and after irrigation with drainage water, it became 40.65%; therefore, there was a slight increase in the content of physical clay. Almost similar changes occurred in soils where irrigation was carried out with collector water. Nevertheless, under the influence of irrigation with mineralized water sources, slight changes in the mechanical compositions of the soils of the upper horizons occurred. Obviously, these changes were associated with the partial dispersion of particles of mechanical fractions under the influence of sodium, which is contained in mineralized irrigation water sources, which were applied during the three years of the growing season. The particle size distribution is associated with the volumetric and specific gravity, as well as the soil porosity. According to a soil science textbook, the general physical properties include the soil density and solid density, as well as the porosity.

The determination of the humus, total nitrogen, and phosphorus contents in our experiment was carried out in three periods in the spring and fall of 2016, as well as in the fall of 2018. There cannot be major changes in the content of humus for a short period of observation, but some changes were recorded. Thus, in the arable horizons of the studied meadow saz soils with a medium loamy mechanical composition, over a three-year period, there was a slight decrease in the content of soil humus in the spring of 2016. In the arable and subarable horizons, the humus content range was 1.131–1.045% (Table 3).

According to the results of statistical tests (MANOVA (dependent variables: humus, nitrogen, all forms of potassium and phosphorus)), there were significant changes in the polychemical composition of the soil, depending on both the time of exposure (years) and exposure options (variants or treatment).

In the second variant, the humus contents in the arable and subsurface horizons equaled 1.129% and 1.10%, respectively, in the third year of the study in the fall of 2018. In the first variant, where for three years the wheat was irrigated with river water, the humus contents in the arable and subsurface horizons reached 1.23 and 1.11%, respectively; that is, there were practically insignificant increases. This corresponds well with the previously published data [21,22].

However, in the second variant, where for three years the wheat was watered with mineralized water with a mineralization rate of 4.2 g/L, as mentioned above, a decrease in humus is observed, both in comparison with the spring of 2016 and in comparison with the studied variants from 2018.

In the arable layer the humus content equaled 1.110%, and in the spring of 2016 it equaled 1.129%. In the subarable horizon over the same period, the change was in favor of a decrease in humus in the variant involving irrigation with drainage water, where the humus content equaled 1.00%, while in the spring of 2016 it was 1.100%. In the first variant, where the irrigation was carried out with river water sources in fall, the humus content in the arable layer was 1.233% and in the subsurface horizon 1.110%, while in the 2nd variant in the subsurface horizon the content equaled 1.00%.

In the other treatments, i.e., in treatments 3 and 4, where the wheat was watered with mineralized water but at a lower concentration, compared to the second treatment, similar changes occurred but less intensively.

At the same time, it can be assumed that in this case there was a migration from the upper horizons to the lower ones, where very weak streaks were observed in the genetic soil profile. This process was more pronounced in the second variant, where the irrigation was carried out with mineralized water sources—with a chloride-sulfate type of mineralization water with a solid residue content of about 4.2 g/L (Dep.2 fall 2018). In our experiment, we did not measure the water discharged for evapotranspiration, as was done in previous research [26]. To assess the water balance and salt concentration in the irrigation water sources and soils more adequately, it is necessary to take into account the main items of arrival and moisture consumption [26,29].

During the indicated period (from spring 2016 to fall 2018), the nitrogen content decreased. As for the amount of gross phosphorus, there were very slight decreases in the arable horizons under treatment 2 in the fall periods of 2016 and 2018. These were associated with an increase in mobile phosphorus under the influence of mineralized water sources and further mixing from the plow horizon to the subplow horizon. Slight increases in gross potassium occurred in the variant irrigated with mineralized water, both in fall 2016 and in fall 2018 (Table 3).

As shown in Figure 2, the changes in agrochemical properties at different soil depths occur differently. This is especially characteristic of gross and mobile forms of potassium and phosphorus.

As for the ratio of carbon and nitrogen, i.e., enrichment with nitrogen, in the initial state this indicator in all horizons of all variants of the experiment varied within the range of 6.7–7.8. As expected, a slight enrichment in nitrogen corresponded to the upper arable horizons. In connection with the changes in the content of humus, this ratio changed to a certain extent. In the fall of 2016, in the variant with river irrigation (treatment 1), in the arable layer it was 6.8 and in the subarable layer it was7.1, while in the second variant the ratios were 7.1 and 6.4; that is, there was still a slight increase in the arable layer in the ratio, while in the subarable layer it declined.

In the fall of 2018, there was an increase in the ratio in the second treatment compared to the first one. In the remaining treatments (3 and 4), similar changes occurred as in the second. According to the grouping of irrigated soils according to mobile forms of nutrients, the soils studied by us before the field experiment (in the spring of 2016) were assessed as very poorly supplied in terms of the content of nitrate–nitrogen (Table 3).

In the fall of 2018, according to the availability of Na-NO_3_, the arable horizons for all treatments were assessed as poorly provided, with the N-NO_3_ contents in the range of 28.1–28.6 mg/kg, while in the arable horizons they remained at the same level as in the spring of 2016, whereby the contents in all subarable horizons of the studied soil variants ranged from 9.2–11.2 mg/kg.

As for the variants treated with irrigation with river and mineralized water sources in fall 2016 and fall 2018, there was no significant difference. However, in the second version of the experiment, if the content range of N-NO_3_ in the plow and subplow horizons was 28.6–11.2 mg/kg, in the treatment 1 it is 28.5–11.10; therefore, there was a very small increase in nitrates in variants irrigated with mineralized water, which was associated with the introduction of nitrates to the mineralized irrigation water. Thus, future investigations are need for clarification of the nitrification and ammonification processes in arid soil under the various practices of irrigation [30].

Nevertheless, in our opinion, an interesting phenomenon was observed regarding the changes in the mobile forms of phosphorus in the soils under the influence of irrigation with river and mineralized water sources. In general, before the start of the experiment, the soils were low in terms of the contents of mobile phosphorus, but in the variants irrigated with mineralized water, as expected, there was a slight change in soil mobile phosphorus in the direction of an increase in the plow horizon before the experiment from 2016. In the spring the mobile phosphorus content was. 22.5 mg/kg and in the fall of 2016 it was 30.6, while in 2018 it was 34.7 mg/kg for the variant irrigated with river water. Similarly, a small increase occurred in the subsurface horizon. With regard to these increases, they were associated with the introduction of mineral phosphorus and the quality of the irrigation water.

It should be emphasized that between treatments 1, 2, 3, and 4, there were also differences in the contents of mobile phosphorus in the soils. In the fall of 2016, in the second variant (irrigated with mineralized water), mobile phosphorus was found in the plow horizon in the amount of 32.7 mg/kg, while in the fall of 2018 the amount was 35.6 mg/kg when it was contained in the same horizons as for the variant irrigated river water (30.6–34.7 mg/kg).

This means that under the influence of the irrigation of wheat with mineralized water, there was a slight increase in mobile phosphorus in the soils, which was associated with the magnesium–sodium composition of the irrigation water. In the subsequent variants, a similar pattern was observed, but it was less pronounced. In the subarable horizons, there was also a slight increase in mobile phosphorus in the variants irrigated with mineralized water. The explanation for this fact is not final, due to the fact that the soils were saline and contained quite a high content of sulfate salts of magnesium and sodium, as well as sodium chloride. Additional research is required in this direction. According to the theory, the magnesium cation quickly binds the phosphorus anions; therefore, it translates into a stationary state. The soils and soil solutions contained numerous cations of different names and geneses, as well as with differing properties. Their complex effect on the content of mobile forms of phosphorus cannot be ruled out. According to the content of mobile potassium, the studied soils belonged to the group of soils with a low degree of availability. The contents in the soils studied by us ranged from 101.5 to 178.8 mg/kg.

It should be especially emphasized that in field conditions, taking into account soil salinity, we took soil samples to determine the mobile forms of microelements. As a result of the analyses, we found that with increasing soil salinity, the amount of molybdenum increases, and in saline soils, and especially in moderately saline soils, a molybdenum-elevated pedogeochemical area forms, with a concentration coefficient range of 6.12–6.67 versus 2.33–2.71 in non-saline and slightly saline soils.

Irrigation with mineralized water sources does not affect the soil salt composition. However, the available data show that in the variants irrigated with mineralized water, a slight increase occurs, which is associated with the introduction of mineralized irrigation water and the weathering of potassium-containing minerals under the influence of these water sources of the above composition. The average mineralization range for drainage, collector–drainage, and mixed water sources was from 2.85 to 4.80 g/L (Table 4). These sources are dangerous in terms of soil salinity, and they belong to the brackish group, while their mineralization rate is estimated below at 5 g/L. In all years of the study (2016–2018), we carried out 3 vegetation irrigation phases, for which the quantities and qualities were close. Taking into account this situation, we averaged these data for each growing season by year, from which it is clear that in these water sources there is no normal soda. In ditch water sources, the indicators of both anions and cations are low compared to mineralized water sources. For example, for hydrocarbonates, the contents in ditch water sources range from 0.0055 to 0.072 g/l, while in mineralized water sources the range is 0.220–0.261 g/L. A similar situation is observed for chlorine, sulfates, and also cations. It was expected that high rates would be characteristic of sulfates, which are contained in mineralized water sources in the range of 1.30–1.72 g/L.

According to the results of the statistical tests (MANOVA), there were insignificant changes in the polychemical compositions of the saline water sources over time (years), but the water sources themselves differed significantly in their compositions from each other (water type).

Depending on the mineralization and qualitative composition of ions, as well as the salts, it should be emphasized separately that the contents of magnesium were close to those of calcium. The contents of magnesium varied within the range of 0.165–0.250 g/L, while calcium was in the range of 0.258–0.320 g/L. During the observation period in spring 2016, i.e., before irrigation with water sources with varying degrees on mineralization, and in fall 2018, after irrigation with mineralized river water in the 0–100 cm soil horizon, the following changes occurred regarding the total mass of salts (Figure 2).

The given data show that the greatest accumulation of salts, as expected, occurred in treatment 2, where the irrigation was carried out with mineralized water with a concentration range of 4.2–4.8 g/L. Additionally, there was a variant with mixed water, for which the salt concentration was about 3.8, as well as treatments 4 and 3. It should be especially noted that in the variant irrigated with river water sources, the accumulation of salts of the order of 17.8 t/ha also occurred. These changes can be more clearly seen in Figure 3, which shows the changes in these salts for spring 2016 and fall 2018. The data in the figure show that the dense residue in the variant with river water irrigation from spring 2016 to fall changed, in the range of 131.5–129.8 t/ha.

In the variant irrigated with mineralized drainage water, this indicator changed very significantly over three years of irrigation, i.e., it fluctuated in the range of 149.3–187.9 t/ha. Over three years in treatment 1, the salts accumulated to the amount of 17.7 t/ha, while in the second treatment this amount was 54.16 t/ha; that is, the accumulation of salts was almost three times greater. In the third treatment, the amount of accumulated salts for the three years of irrigation with mineralized water was 51.9 t/ha; compared to the second treatment, this is 2.3 t/ha less, while compared to treatment 1 (irrigation with ditch water), this is three times more. In the treatment with mixed water, the accumulation of salts was 2.4 times greater compared to the first treatment.

The given data indicate that the use of mineralized water sources with a mineralization range of 3.8–4.2 g/L, even in medium loamy soils, leads to an increase in soil salinity. Soil salinization is typical even for chernozems, for which the irrigation of agricultural crops with river water sources in the conditions of the Krasnodar Territory [30] led to the salinization of southern chernozems. The irrigation of chernozems also leads to increases in the volumetric mass and exchangeable sodium and the water-soluble salts in soils, as well as decreases in the yield and quality of agricultural products obtained from these soils. The use of mineralized water sources for irrigation has even more significant negative impacts not only on the soils, but also on the ecology and geochemistry of the landscape. At the same time, the greater the mineralization of the irrigation water, the more it will be required to maintain the leaching water regime, meaning drainage will be required and salinization and soil degradation will become more likely.

In general, from the given data, it can be seen that the amounts of salts during irrigation increased from year to year, especially in the upper 0–100 cm soil layer. The most intensive accumulation occurred in treatments 2 and 3. Thus, despite such a significant introduction of water-soluble salts, the soils remained moderately saline and the arable layers remained slightly saline.

The mathematical processing of the obtained data for the dense residues, where the contents (remnants of salts after evaporation in laboratory conditions) in tons per hectare were obtained for the 0–100 cm soil layer, was characterized by the following indicators: the average content was 114.4 t/ha, the standard deviation was ±56.1, and the coefficient of variation was 49.0. The data obtained were well correlated with the average wheat yield of 41.65 c/ha, for which the correlation coefficient was positive and amounted to 0.51. Studying the possibility of using mineralized water sources for irrigation, crops are obtained by irrigating the soil with water sources with various concentrations of salts. It is necessary in each individual case to study the soil and agrobiological processes under specific soil and climatic conditions. Under conditions similar to ours in loamy soils, the possibility of using mineralized water for irrigation is limited by a number of factors, such as the level of groundwater, the mechanical composition, the chemical state of the soils, the plant composition, and other factors.

For the different conditions, the maximum permissible concentrations of salts in the mineralized water sources are different. As the groundwater level rises, the possibility of irrigating plants with mineralized water sources decreases. Under these conditions, the salts washed out during irrigation from the arable layer are again returned by capillary movements from the groundwater. In field conditions, the creation of a leaching regime is associated with the irrigation rate, meaning a leaching irrigation regime during the growing season is not always justified. This is due to the leaching of nutrients and water-soluble humus, which nevertheless leads to a certain decrease in the yield of the agricultural crops due to a violation of the nutritional regime of the plants and the physicochemical properties of soils. In favor of the use of mineralized water sources with increased mineralization, one can cite the data from foreign authors, as given by Stroganov et al. [36], who showed that sweet corn, flax, and cotton showed increased yields with sulfate mineralization. The same author stated that the role of sulfur in the normal metabolism is well known. It is of paramount importance in the life of plants and is an integral part of many components of the cell, playing important roles in the properties and structural transformations of the protein molecule, in redox processes, and in the energy metabolism of the cell.

Vladychensky also noted that contents of soda and chlorides over 1–2 g/L make water completely unsuitable for irrigation in the chemical sense. However, nowadays, it is very hard to find such water in river sources. Additionally, practically agriculture involves use secondary water sources containing essential salts. According to Gabali, the water of the Nile in relation to the chemical composition undergoes monthly changes. When studying the mineralization of groundwater in the zone of action of the Nile and beyond, it was found that with an increase in the concentration of salts in the groundwater of the delta, the amount of HCO_3_ decreases and the amounts of SO_4_ and Cl increase. The same author pointed out that due to the apparent lack of water in some areas, water sources with a salinity range of 2–11 g/L have been used for irrigation over the past 30 years [18]. In the study area containing gypsum soils, on the contrary, the decrease in soil fertility during irrigation and flushing with mineralized water occurs mainly due to salinization and not due to soil solonetzization. Yuldashev, having carried out leaching irrigation experiments in the saline soils of the Ferghana Regional Experimental Station, came to the conclusion that after leaching the soils with groundwater, the content of phosphoric acid does not change significantly. There is no accumulation of phosphorus in the plow horizon of the soil, but on the contrary, the phosphorus partially moves down the profile. Using drainage water with a dense residue range of 2.5–3.1 for irrigation, it is possible to conclude that water with contents of up to 4 g/L can be used for irrigating cotton in light loamy soils, which can be successfully used for washing highly saline land areas. In this case, irrigation should be carried out against a background of a well-functioning drainage system [36,37]. Due to the rapid drying of the soil in zones of intense wind activity, such as Central Fergana, it is necessary to apply irrigation rates of the order of 1000–1200 m^3^/ha or more, or irrigation rates of 4000–6000 m^3^/ha. At 100 tons/ha, both inputs resulted in the highest values for all investigated traits. It was also found that the yield and yield components that were obtained from bagasse ash treatments overwhelmed those from filter cake treatments, except in terms of the tiller, dry biomass, and straw yield values. The linear regression analysis revealed a significant and positive relationship between grain yield and the total N, P, K, S, Ca, Mg, Cu, and Zn uptake. A linear relationship between the grain yield and N and Zn uptake was found, while the association between grain yield and the total P, K, S, Ca, Mg, and Cu uptake was quadratic [38].

In general, along with other things, plants need almost all ions of the water extract in one quantity or another. These ions play various metabolic roles in plants. For example, potassium and sodium change the activity of enzymes, and chlorine is involved in photosynthetic activity, as well as in creating a crop. In the presence of high sulfur concentrations, there is a decrease in the biosynthesis of sulfur-containing amino acids and the incorporation of sulfur into proteins. Sulfur deficiency also affects the formation of sulfur-containing amino acids, including proteins. The water sources studied by us, which were used for irrigating wheat of the Polovchanka variety, in terms of their general mineralization were weakly mineralized, whereby the contents of the dense residue varied in the range of 2.8–4.8 g/L. In terms of their anionic composition regarding sulfate mineralization, the Cl:SO_4_ ratios fluctuated in the range of 0.12–0.19. The effects of irrigation on phenological indicators and wheat yields are given below (Table 5).

From the data presented in the table, it can be seen that mineralized water sources with varying degrees of salinity affect the growth, development, and yield of wheat in different ways. For example, the greatest plant growth was observed for the fourth treatment, for which the heights of the wheat plants in the maturity phase reached 110.4–114.3 cm. Approximately the same pattern was observed in the previous growth phases. The second place in terms of these indicators was taken by the treatment where irrigation was carried out only with collector water. In all years of the study, three irrigation phases were carried out with the above concentrations of irrigation water sources. It was expected that irrigation with mineralized water would definitely reduce the wheat and cotton yields. Despite our expectations, it turned out that not all mineralized water sources adversely affect the yields of wheat. We found that the amounts of wheat grain and straw under the influence of irrigation with mixed collector–drainage and river water sources with mineralization rates of up to 3.8 g/L did not decrease the yields of Polovchanka wheat, but instead increased it compared to the control, where only ditch (river) water sources with concentrations of water-soluble salts of up to 1.1 g/L were used. It can be assumed that when wheat is irrigated with mineralized water, the amount of water-soluble sulfur in the soil will increase, which is included here in the composition of plant nutrients. Thus, the ditch water taken for irrigation practices directly from the source (used as the control) is not the best in terms of the yield rate. Nevertheless, ditch water sources will inevitably be involved in agriculture as more saline water sources.

Consequently, the nutritional status of wheat in relation to sulfur is improved. In addition, it is possible that the positive effects of irrigation with mineralized water on the growth and development, as well as the yield of Polovchanka wheat, were due to the relative balance of ions in the irrigation water sources. In general, for three years, the wheat yield (Table 3) for the variant irrigated with mixed water was 44.5 c/ha, while for the control variant it was 43.1 c/ha. Consequently, the placement of agricultural crops and the yield determine the admissibility of the concentration in the irrigation water, on which the quality of the wheat also depends. However, it should be especially emphasized that in the variant irrigated only with drainage water, the yield decreased and amounted to 37.9 c/ha.

The mathematical and statistical processing of the obtained data showed that with an average yield (on average for 3 years according to the experiment) of 41.65 c/ha, the standard deviation is ±2.8, the coefficient of variation is ±6.9, and the correlation coefficient for the dense soil residue (the average in the 0–100 cm layer over three years 1.21%) is 0.58; that is, the relationship is positive for the average tightness.

The data given in Table 6 show that the cultivation of wheat under conditions of irrigation with mineralized water is relatively justified. In the first year (2016) of wheat cultivation, at an average cost of 1 kg of wheat per 680.5 sums, the profitability margin in the variant with ditch water irrigation was 20.6%. In the treatment with irrigation with drainage water 16.5%, in treatment 3 (irrigation with collector water) the profitability margin was 19.2%, while in treatment 4, where irrigation was carried out with mixed water, the profitability margin was 24.4%. With the growth of the yield and price per 1 kg of wheat, the profitability increases. In the second year (2017), the yield rate in the first treatment was 27.1% and in the 4th treatment it was 33.0%. In 2018, in treatment 1, the profitability margin was 39.9%, while in treatment 4, the profitability margin was 38.5%. For the second and third treatments, the profitability margin range was 24.9–31.2%.

The yield rate for irrigation with drainage water was 19.5%, for treatment 1 (irrigation with ditch water) it was 29.2%, and for the rest the yield range was 26.2–31.9%. It should be emphasized that in addition to pure economic efficiency, there is an ecological efficiency, which consists of saving clean river water to the amount of 2400 m^3^/ha annually, the properties of which must be preserved and improved. This position is of no small importance in the national economy.

## 4. Conclusions

The results of the analysis of the water extracts made it possible to establish that the irrigated meadow saz soils, after a three-year influence of irrigation with ditches and mineralized water sources, remained in the group of moderately saline soils, starting from the subarable horizons, while the arable horizons remained weakly saline.

Despite the rather high rates of sulfate salts in the studied soil variants, soda formation did not occur, due to the insufficient amounts of carbon organic matter. Of the toxic salts, the highest contents of Na_2_SO_4_ were found in variants 2, 3, and 4, followed by MgSO_4_ and NaCl. In the second variant, the contents of toxic salts fluctuated in the range of 0.292–0.803% and were on the border of the results for very highly saline soils. The total mass of salts in the 0–100 cm soil layer correlated with the yield of wheat of the Polovchanka variety, the coefficient of which was 0.5, with a wheat yield of 41.65 c/ha, while the total content in the layer equaled 114.5 t/ha.

Under the influence of irrigation with mineralized water, a slight leaching of humus from the upper horizons into the lower layers occurs. The ratio of carbon to nitrogen in the soil from the top to bottom is not enriched by nitrogen in all horizons, and varies from 6.7 to 7.8. The highest nitrogen enrichment occurs in the arable horizons. According to the contents of the mobile forms N, P, K, the soils were poorly supplied; after a series of irrigation phases, they remained the same. However, there were signs of the growth of mobile phosphorus in the variants watered with mineralized water sources. There were relatively greater contents of magnesium salts in the saline horizons than calcium in the secondary saline soil. Calcium and potassium, under the influence of a constant migration flow of elements under these conditions, were gradually replaced by magnesium, especially in the upper area and in contact with groundwater horizons, while magnesium passed into a non-exchange state.

The mineralization rates for both the drainage and collector–drainage mixed water sources of the region ranged from 2.85 to 4.8 g/L, making them dangerous in terms of soil salinity and meaning they belong to the brackish group; their mineralization is estimated at below 5 g/L. According to the ratio of chlorine and sulfuric acid ions, these sources were characterized as being dominated by sulfates. The SAR interval ratios, ranging from 0.77 to 2.09, indicated their salinity.

The irrigation of wheat with water sources of various compositions for 3 years yielded a grain yield range of about 37.9–44.5 c/ha. In the treatment involving irrigation only with river water the yield was 43.1 c/ha, while in the second treatment, where the irrigation was carried out only using drainage water, the yield of grains was 37.9 c/ha; that is, there were clear negative effects of the salinity and chemical composition of the drainage water, whereby the profitability was also low, amounting to 19.5%. In the subsequent treatments (3, 4), this indicator varied within the range of 26.2–3.9%. The calculations of the economic efficiency of the cultivation of wheat of the Polovchanka variety when irrigated with mineralized water confirmed the feasibility and profitability of this method.

## Figures and Tables

**Figure 1 plants-11-03291-f001:**
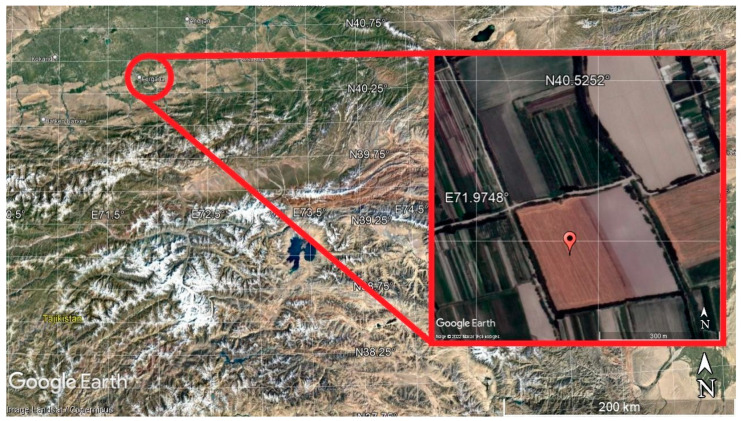
Location of the study area and the soil section of the irrigated meadow saz soil in the image from the Google Earth service.

**Figure 2 plants-11-03291-f002:**
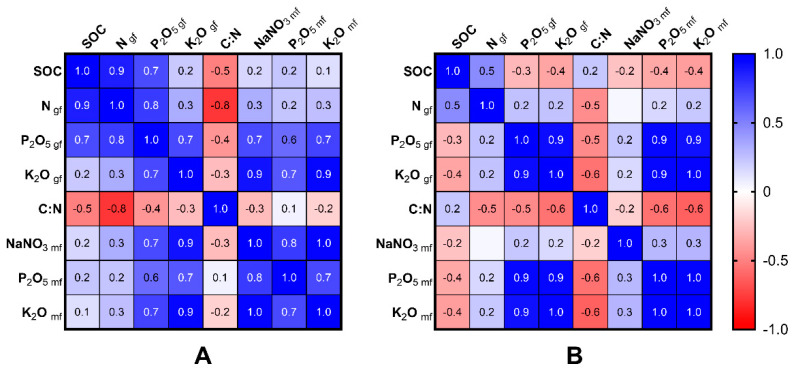
Spearman correlation matrix between agrochemical characteristics: (**A**) 0–32 cm depths (*n* = 12); (**B**) 32–50 cm depths (*n* = 12). Note: gf—gross form; mf—mobile form.

**Figure 3 plants-11-03291-f003:**
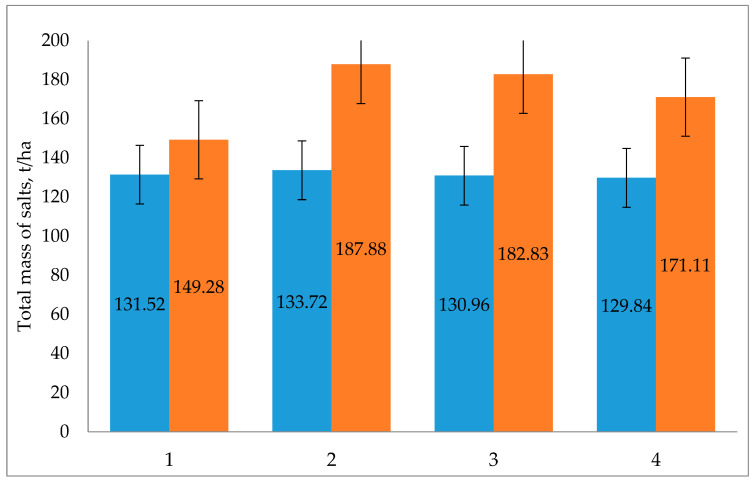
Changes in the total mass of the salts in t/ha in the 0–100 cm soil layer. Row 1: The mass of salts in spring 2016. Row 2: The mass of salts in fall 2018. Note: 1, 2, 3, and 4 represent the treatments.

**Table 1 plants-11-03291-t001:** Toxicity levels of dominant salts.

Salt	Na_2_CO_3_	NaCl	NaHCO_3_	Na_2_SO_4_
Degree of toxicity	10	3	3	1

**Table 2 plants-11-03291-t002:** Particle size distribution of the soils.

Treatments	Depth, cm	Fractions, mm							Physical Clay, <0.01
		>0.25	0.25–0.1	0.1–0.05	0.05–0.01	0.01–0.005	0.005–0.001	<0.001	
2016									
Initial state	0–32	0.70	3.15	8.51	47.29	9.30	15.20	15.85	40.35
	32–50	3.15	6.20	11.10	47.00	7.30	19.10	6.15	32.55
	50–70	7.80	6.70	6.11	4870	9.15	16.14	5.40	30.69
	70–90	9.15	10.41	10.10	39.13	13.41	13.10	4.70	31.21
	90–110	10.10	10.06	10.01	37.98	13.40	15.38	3.15	31.85
	110–130	10.50	10.90	8.30	39.81	11.60	13.74	5.15	30.49
2018									
1	0–32	0.72	3.10	8.50	47.03	9.30	15.30	16.05	40.65
	32–50	3.05	6.10	11.15	46.9	7.40	19.20	6.20	32.80
	50–70	7.90	6.80	7.10	47.35	9.20	16.20	5.45	30.85
	70–90	9.45	10.51	10.15	39.19	12.90	13.20	4.60	30.70
	90–110	10.20	10.11	9.05	39.04	13.10	15.20	3.30	31.60
	110–130	11.10	11.10	9.05	37.84	12.1	13.54	5.30	30.91
2	0–32	0.72	3.10	8.45	47.18	9.30	15.25	16.00	40.55
	32–50	3.10	6.15	11.10	46.8	7.35	19.20	6.30	32.85
	50–70	7.80	6.75	6.27	48.48	9.10	16.20	5.40	30.70
	70–90	9.30	10.50	10.10	39.14	13.01	13.30	4.65	30.96
	90–110	10.10	10.20	10.00	37.95	13.30	15.25	3.20	31.75
	110–130	10.30	11.20	8.45	39.19	12.0	13.64	5.20	30.86
Min		0.7	3.1	6.11	37.84	7.3	13.1	3.15	30.49
Max		11.1	11.2	11.15	48.7	13.41	19.2	16.05	40.65
Mean		7.0	7.9	9.1	43.1	10.7	15.5	6.8	32.9
CV		55.33%	36.95%	16.91%	10.39%	21.37%	12.98%	63.87%	10.87%

**Table 3 plants-11-03291-t003:** Agrochemical characteristics of soils.

Variants of Treatment	Depth, cm	Humus, %	Bulk Forms, %			C:N	Mobile Forms, mg/kg		
			N	P_2_O_5_	K_2_O		NaNO_3_	P_2_O_5_	K_2_O
Spring—2016									
1	0–32	1.13	0.111	0.255	1.839	6.7	1.91	22.5	145.5
	32–50	1.08	0.098	0.245	1.109	7.2	19.2	15.8	101.2
2	0–32	1.12	0.095	0.26	1.841	7.8	18.2	22.1	150.3
	32–50	1.1	0.095	0.25	1.09	7.6	18.1	16.2	101.5
3	0–32	1.13	0.11	0.26	1.791	6.8	19.2	22.7	148.5
	32–50	1.07	0.096	0.245	1.111	7.3		15.9	105.6
4	0–32	1.13	0.111	0.255	1.803	6.7	18.8	22.6	146
	32–50	1.09	0.094	0.25	1.11	7.7	8.9	15.9	101
Fall—2016									
1	0–32	1.24	0.121	0.277	1.904	6.8	28.1	30.6	165.5
	32–50	1.1	0.101	0.265	1.121	7.1	10.1	20.8	110.2
2	0–32	1.13	0.103	0.267	1.909	7.1	28.25	32.7	170.6
	32–50	1.06	0.087	0.271	1.128	6.4	10.2	21.8	115.1
3	0–32	1.15	0.113	0.269	1.9	6.8	28.15	32.6	170.5
	32–50	1.11	0.098	0.277	1.129	7.4	10.15	21.7	115.05
4	0–32	1.18	0.115	0.268	1.909	6.7	28.12	31.5	170.4
	32–50	1.09	0.101	0.276	1.124	7.1	10.1	21.6	115
Fall—2018									
1	0–32	1.23	0.127	0.281	2.003	6.3	28.5	34.7	175.7
	32–50	1.11	0.112	0.277	1.224	6.2	11.1	23.5	125.5
2	0–32	1.11	0.107	0.268	2.025	6.9	28.6	35.6	180.9
	32–50	1	0.1	0.281	1.235	6.6	11.2	24.5	126.5
3	0–32	1.14	0.118	0.278	2.014	6.3	28.5	25.5	178.8
	32–50	1.07	0.095	0.28	1.229	7.3	11.18	24	126
4	0–32	1.17	0.119	0.274	2.01	5.9	28.55	24.8	176.5
	32–50	0.99	0.097	0.287	1.226	6.7	11.15	23.6	125.7

*MANOVA results: Years—df = 16, F = 7.81, p < 0.005 (significant); Variants or treatment—df = 24, F = 1.83, p < 0.005 (significant).*

**Table 4 plants-11-03291-t004:** Average chemical compositions of irrigation water sources (g/L).

Water Type	Dry Remnant	CO_3_^2−^	HCO_3_^−^	Cl^−^	SO_4_^2−^	Ca^2+^	Mg^2+^	K^+^	Na^+^	mg/L		
										NH_4_^+^	NO_3_^−^	P_2_O_5_
2016												
Ditch	0.925	No	0.066	0.09	0.555	0.115	0.058	0.031	0.041	10	9	3.5
Drainage	4.21	Track.	0.252	0.22	1.55	0.275	0.24	0.09	0.141	15.1	14.1	4.9
Collector	2.85	No	0.22	0.12	1.305	0.258	0.165	0.071	0.131	10.2	8.15	2.5
Mixed	3.501	No	0.23	0.15	1.438	0.265	0.201	0.8	0.135	12.1	10.15	2.7
2017												
Ditch	1.105	No	0.072	0.08	0.63	0.12	0.068	0.035	0.051	11	8.5	3
Drainage	4.8	Track.	0.261	0.23	1.624	0.285	0.25	0.085	0.162	14.1	19.1	4
Collector	3.42	No	0.24	0.18	1.458	0.31	0.17	0.07	0.184	11.2	9.1	2.8
Mixed	2.81	No	0.22	0.14	1.544	0.265	0.211	0.072	0.152	12.1	8.5	3
2018												
Ditch	0.845	No	0.055	0.09	0.648	0.12	0.071	0.024	0.05	10	8	2.5
Drainage	4.02	Track.	0.24	0.235	1.718	0.28	0.245	0.084	0.172	13.1	19	3.5
Collector	2.9	No	0.23	0.2	1.444	0.32	0.18	0.061	0.155	10.2	10	3
Mixed	3.25	No	0.22	0.168	1.614	0.29	0.22	0.072	0.161	12.3	9	2.8

*MANOVA results: Years—df = 14, F = 3.45, p = 0.24 (insignificant); Water type—df = 21, F = 30.66, p < 0.005 (significant).*

**Table 5 plants-11-03291-t005:** Growth, development, and weight values of 1000 wheat seeds.

Treatments	Mineralization, g/L	Plant Growth, cm				Weight 1000 Seeds, g.
		Tubing Stage	Earing	Milky Wax Ripeness	Maturity	
2016						
1	0.92	81.0	35.0	97.5	100.1	42.8
2	4.20	84.5	33.3	98.8	101.2	42.2
3	2.80	90.5	103.0	105.3	110.4	42.1
4	3.50	98.4	111.9	108.4	114.3	42.7
2017						
1	1.10	79	90.0	92.5	98.1	41.8
2	4.80	81.5	90.1	94.4	100.3	42.0
3	3.42	90.1	95.8	99.4	101.4	42.1
4	3.80	94.3	101.9	103.0	110.1	42.8
2018						
1	0.85	96.3	110.7	107.4	110.4	42.6
2	4.02	85.5	95.6	95.5	92.3	42.0
3	2.90	93.3	96.5	98.4	102.3	42.8
4	3.25	95.4	110.9	105.4	111.2	42.8

**Table 6 plants-11-03291-t006:** Wheat grain yields (c/ha) for 3 years (above the line—grain; below the line—straw).

Treatments	2016	2017	2018	Mean
1	$\frac{43.0}{58.2}$	$\frac{41.0}{56.3}$	$\frac{45.3}{68.7}$	$\frac{43.1}{61.1}$
2	$\frac{40.8}{58.8}$	$\frac{36.5}{49.3}$	$\frac{36.3}{58.4}$	$\frac{37.9}{55.5}$
3	$\frac{42.2}{66.5}$	$\frac{41.6}{62.4}$	$\frac{39.6}{64.2}$	$\frac{41.2}{64.4}$
4	$\frac{44.9}{64.7}$	$\frac{44.2}{63.8}$	$\frac{44.3}{71.8}$	$\frac{44.5}{66.8}$

*Fisher SLD—±1.07 c/ha.*

## Data Availability

Not applicable.

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
