# Peer review of "Influence of Mineralized Water Sources on the Properties of Calcisol and Yield of Wheat (*Triticum aestivum* L.)"

_plants, 2022, doi:10.3390/plants11233291_

Round 1
Reviewer 1 Report
This manuscript conducted the Influence of mineralized waters on properties of Calcisol and yield of wheat (Triticum aestivum L.). The results was useful and interesting. However, the MS need to be improved before publication.
1. Introduction should be shortened to highlight the importance of this study, some presentation could be moved into the discussion section.
2. Figure 1 should be added the detail information i.e. longitude and latitude.
3. All of the tables need the necessary notes, formats of some tables need to be improved such as table 6.
4. Most data in the tables and figures were lack of the significant analysis.
5. Discussion section was missing.
Author Response
Dear reviwer!
thank forvaluable comments,
all of them were taken into account, namely
- Introduction should be shortened to highlight the importance of this study, some presentation could be moved into the discussion section.
The introduction has been shortened and some material has been moved to Materials and Methods
- Figure 1 should be added the detail information i.e. longitude and latitude.
corrected
- All of the tables need the necessary notes, formats of some tables need to be improved such as table 6
corrected
- Most data in the tables and figures were lack of the significant analysis.
Corrected, additional statistic provided
- Discussion section was missing
The discussion is presented in the combined results and discussion section
Reviewer 2 Report
The article describes the results of field experiment, when soil plots were irrigated by four types of water with different mineralization. The broad spectrum of soil properties and some characteristics of wheat crop were measured.
The introduction part of the article describes the aspects of crop irrigation with mineralized water. The review of the subject is quite detailed. However, there are some parts that could be improved, especially by language check
The materials and methods section needs to be improved. Please clarify when sampling was performed, what type of samples were taken and how many. Were soil profiles studied? if yes, how many?
Results and discussion section is more or less detailed. However, it definitely will benefit from the language check.
Conclusions are the repetition of results. I would recommend to shorten this section.
Line 14 "conservation agriculture (SWC)" I suggest to place the definition for the abbreviation here to avoid confusion further
Line 15 required - I suggest to change it to "necessary" or "needed"
Line 16 topical is also not used
Line 22 "were watered" is better change to "were applied"
Line 23-27 - it would be better to split the sentences
Line 36 - "degradation of land and water resources is growing" suggest the change to "growing degradation of land and water resources"
Line 42-47, 56-59 - please split the sentence to make it more clear
Line 49-54 - (major change) seems that this part is not relevant to the article topic and should be removed
Line 60, 61 - repeat of "horticultural"
Line 98 "no higher than"
Line 119 "soils and soils" please remove the repeat
Line 120-122 please make this sentence more clear
Line 160-162 - please describe where results were obtained
Line 163-164 - the sentence is unclear
Line 177 should it be regions?
Line 187 - should it be returned?
Lines 225-235 - (major change) seems that this part about N application is not relevant to the article topic and should be removed
Line 236 - data presented by whom?
Line 256-259 - (major change) this part is irrelevant to the article and should be removed
Line 260-263 seems that a part of sentence was lost. "Over the past years, ..." - did what?
line 264-265 - this statement is not supported with facts
Line 266-270 - (major change) the purpose of the study should be formulated clearly. Did authors study the influence of two types of irrigation water (mineralized collector-drainage water and collector-drainage mixed water)? What was under the influence (soil formation, wheat productivity)?
Were the effects studied for only one soil?
Line 271-272 - "determine the degree of mineralization of waters by the danger of their salinization and alkalinization of irrigated soils" please change the sentence (it seems that salinization and alkalinization happens to waters and not to soils)
Line 280, Objects of study - I would recommend to move this section to Materials and methods
Line 294 what was meant by accounting area?
Line 298 drawings? what was meant?
Line 300-301 sentence seems unfinished. Please check.
Line 303 - please add m3/ha
Line 307 - what methodology?
Line 308 - "in accordance with the program" - what program?
Line 311-312 - what type of agricultural treatment was applied?
Line 315 - humus content
Line 314-323 - please provide references for methods used
Line 331 - the Table 2 does not contain the described information (it's about soil particles distribution)
Line 337 - "It was founded in 2016" should it be "soil profile was dug and described in 2016?
Line 355-357 - this part seems to be irrelevant to other article or placed in a wrong place and should be removed
Line 358-361 - I would recommend to split the sentence
Line 362 please check what was meant
Line 364 " taste" should be replaced by seemed
Line 384-390 - (major change) this part is irrelevant to the article and should be removed
Line 415-422 - humus or SOC? there is SOC in the table 3
Line 455, 477, 505 - please unify the names of sampling times, it should be autumn instead of fall
Line 525 - what is KK?
Line 527 - contents of what?
Line 567 Figure 2 Row 3 is not necessary to show
Line 598-600 content of what? this part should be described in more detail
Line 638 Please check terms in the Table 5, Trubko-ing is not correct
Line 647-649 - the fact about the reduction of yields is mentioned, but the control is not clarified. Yields are reduced in case of mineral water irrigation compared to what (yields without irrigation, or with irrigation with fresh water)?
Line 688 (major change) how profitability was calculated? this should be described in methods
Line 698-702 - repetition of results, please check spelling
Author Response
Review 1
The article describes the results of field experiment, when soil plots were irrigated by four types of water with different mineralization. The broad spectrum of soil properties and some characteristics of wheat crop were measured.
The introduction part of the article describes the aspects of crop irrigation with mineralized water. The review of the subject is quite detailed. However, there are some parts that could be improved, especially by language check
The materials and methods section needs to be improved. Please clarify when sampling was performed, what type of samples were taken and how many. Were soil profiles studied? if yes, how many?
Corrected, clarified
Results and discussion section is more or less detailed. However, it definitely will benefit from the language check.
Conclusions are the repetition of results. I would recommend to shorten this section.
Line 14 "conservation agriculture (SWC)" I suggest to place the definition for the abbreviation here to avoid confusion further
Corrected
Line 15 required - I suggest to change it to "necessary" or "needed"
Changed
Line 16 topical is also not used
Corrected
Line 22 "were watered" is better change to "were applied"
Corrected
Line 23-27 - it would be better to split the sentences
Corrected
Line 36 - "degradation of land and water resources is growing" suggest the change to "growing degradation of land and water resources"
Chenged
Line 42-47, 56-59 - please split the sentence to make it more clear
Clarified
Line 49-54 - (major change) seems that this part is not relevant to the article topic and should be removed
Removed
Line 60, 61 - repeat of "horticultural"
Corrected
Line 98 "no higher than"
Corrected
Line 119 "soils and soils" please remove the repeat
Corrected
Line 120-122 please make this sentence more clear
Clarified
Line 160-162 - please describe where results were obtained
Added reference
Line 163-164 - the sentence is unclear
Sentence was rewrite
Line 177 should it be regions?
Corrected
Line 187 - should it be returned?
Corrected
Lines 225-235 - (major change) seems that this part about N application is not relevant to the article topic and should be removed
Removed
Line 236 - data presented by whom? – deleted, corrected.
Line 256-259 - (major change) this part is irrelevant to the article and should be removed
removed
Line 260-263 seems that a part of sentence was lost. "Over the past years, ..." - did what?
Corrected
line 264-265 - this statement is not supported with facts
Sentence was corrected
Line 266-270 - (major change) the purpose of the study should be formulated clearly. Did authors study the influence of two types of irrigation water (mineralized collector-drainage water and collector-drainage mixed water)? What was under the influence (soil formation, wheat productivity)?
Were the effects studied for only one soil?
Corrected, clarified.
Line 271-272 - "determine the degree of mineralization of waters by the danger of their salinization and alkalinization of irrigated soils" please change the sentence (it seems that salinization and alkalinization happens to waters and not to soils)
Corrected, clarified.
Line 280, Objects of study - I would recommend to move this section to Materials and methods
Moved
Line 294 what was meant by accounting area?
Corrected the section is removed to below chapter.
Line 298 drawings? what was meant?
Changed - visualization
Line 300-301 sentence seems unfinished. Please check.
Corrected
Line 303 - please add m3/ha
added
Line 307 - what methodology?
Clarified, corrected.
Line 308 - "in accordance with the program" - what program?
Corrected - the design of study
Line 311-312 - what type of agricultural treatment was applied?
corrected
Line 315 - humus content
corrected
Line 314-323 - please provide references for methods used
References was added
Line 331 - the Table 2 does not contain the described information (it's about soil particles distribution)
Description added on lines 331-336, 343 - 353
Line 337 - "It was founded in 2016" should it be "soil profile was dug and described in 2016?
Corrected
Line 355-357 - this part seems to be irrelevant to other article or placed in a wrong place and should be removed
Removed
Line 358-361 - I would recommend to split the sentence
Shortened
Line 362 please check what was meant
Removed
Line 364 " taste" should be replaced by seemed
Corrected
Line 384-390 - (major change) this part is irrelevant to the article and should be removed
Removed
Line 415-422 - humus or SOC? there is SOC in the table 3
Humus
Line 455, 477, 505 - please unify the names of sampling times, it should be autumn instead of fall
Unifyed
Line 525 - what is KK?
clarified
Line 527 - contents of what?
corrected
Line 567 Figure 2 Row 3 is not necessary to show
corrected
Line 598-600 content of what? this part should be described in more detail
Clarified - dense residue (remnant of salts after evaporation in laboratory conditions)
Line 638 Please check terms in the Table 5, Trubko-ing is not correct
Corrected to wheat tubing
Line 647-649 - the fact about the reduction of yields is mentioned, but the control is not clarified. Yields are reduced in case of mineral water irrigation compared to what (yields without irrigation, or with irrigation with fresh water)?
Here the control was ditch irrigation scenario, clarified, corrected.
Line 688 (major change) how profitability was calculated? this should be described in methods
corrected
Line 698-702 - repetition of results, please check spelling
Corrected
Round 2
Reviewer 2 Report
The description of objects and methods, the introduction and results sections were improved. However, I still recommend some changes, as there are some sentences that are formulated unclearly,
Lines 22-27 I recommended splitting the sentence, but now it’s still not clear. It would be better to move the information regarding the experiment to Line18: “The experiment on wheat cultivation on saline soils with irrigation with mineralized water was set for 3 years”. And leave the other part of sentence in Line 22 and further: “Our results showed that in conditions of irrigation by waters with a mineralization of up to 2.8-3.5 g/L, wheat yield increased by 1.5 c/ha compared to the control. The use of mineralized water for irrigation purposes will allow saving clean, river water.”
Lines 42-48 I would suggest the following change for this paragraph:
"For this reason, it is necessary to determine the impact of SWC on agricultural soil fertility, its chemical and geochemical properties in the conditions of irrigation with mineralized water. The irrigation with mineralized waters may help to increase grain yields and to use land and water resources more sustainably. On the globe, the need for water is constantly growing due to the growth and development of the national economy."
Line 254 - please check "experimental"
Line 488 Table 4 should be "MANOVA"
Lines 600-602 Please change the sentence to “It was expected that irrigation with mineralized water would definitely reduce wheat and cotton yields. Despite our expectations, it turned out that not all mineralized waters adversely affect the yield of wheat.”
Line 604 - "3.8 g/L does not decrease"
Line 606 - please add "up to 1.1 g/L was used"
Author Response
Reviewer 2
Dear reviwer! thank you for your suggestions and work with our text, Our comments and replies are given below.
The description of objects and methods, the introduction and results sections were improved. However, I still recommend some changes, as there are some sentences that are formulated unclearly.
Lines 22-27 I recommended splitting the sentence, but now it’s still not clear. It would be better to move the information regarding the experiment to Line18: “The experiment on wheat cultivation on saline soils with irrigation with mineralized water was set for 3 years”. And leave the other part of sentence in Line 22 and further: “Our results showed that in conditions of irrigation by waters with a mineralization of up to 2.8-3.5 g/L, wheat yield increased by 1.5 c/ha compared to the control. The use of mineralized water for irrigation purposes will allow saving clean, river water.”
The abstract has been revised in accordance with the recommendations of
Lines 42-48 I would suggest the following change for this paragraph:
"For this reason, it is necessary to determine the impact of SWC on agricultural soil fertility, its chemical and geochemical properties in the conditions of irrigation with mineralized water. The irrigation with mineralized waters may help to increase grain yields and to use land and water resources more sustainably. On the globe, the need for water is constantly growing due to the growth and development of the national economy."
Corrected
Line 254 - please check "experimental"
Corrected
Line 488 Table 4 should be "MANOVA"
Corrected
Lines 600-602 Please change the sentence to “It was expected that irrigation with mineralized water would definitely reduce wheat and cotton yields. Despite our expectations, it turned out that not all mineralized waters adversely affect the yield of wheat.”
Corrected
Line 604 - "3.8 g/L does not decrease"
Corrected
Line 606 - please add "up to 1.1 g/L was used"
Corrected